# Comparative metabolomics with Metaboseek reveals functions of a conserved fat metabolism pathway in *C. elegans*

Maximilian J. Helf [1,3], Bennett W. Fox [1,3], Alexander B. Artyukhin[2], Ying K. Zhang[1] & Frank C. Schroeder [1✉]

Untargeted metabolomics via high-resolution mass spectrometry can reveal more than 100,000 molecular features in a single sample, many of which may represent unidentified metabolites, posing significant challenges to data analysis. We here introduce Metaboseek, an open-source analysis platform designed for untargeted comparative metabolomics and demonstrate its utility by uncovering biosynthetic functions of a conserved fat metabolism pathway, α-oxidation, using *C. elegans* as a model. Metaboseek integrates modules for molecular feature detection, statistics, molecular formula prediction, and fragmentation analysis, which uncovers more than 200 previously uncharacterized α-oxidation-dependent metabolites in an untargeted comparison of wildtype and α-oxidation-defective *hacl-1* mutants. The identified metabolites support the predicted enzymatic function of HACL-1 and reveal that α-oxidation participates in metabolism of endogenous β-methyl-branched fatty acids and food-derived cyclopropane lipids. Our results showcase compound discovery and feature annotation at scale via untargeted comparative metabolomics applied to a conserved primary metabolic pathway and suggest a model for the metabolism of cyclopropane lipids.

[1] Boyce Thompson Institute and Department of Chemistry and Chemical Biology, Cornell University, Ithaca, NY 14853, USA. [2] Chemistry Department, College of Environmental Science and Forestry, State University of New York, Syracuse, NY 13210, USA. [3] These authors contributed equally: Maximilian J. Helf, Bennett W. Fox. ✉email: fs31@cornell.edu

Widespread adoption of high-resolution mass spectrometry (HRMS) for untargeted metabolomics has revealed a vast universe of biogenic small molecules, including a large number of compounds whose chemical structures have not been elucidated ("unknowns")[1]. Many of these metabolites may serve important biological functions, as intra- or intercellular signaling molecules, e.g., as hormones, or mediating communication at the inter-organismal level, e.g., in host-microbe interactions or as pheromones[2–5]. Their large-scale identification, quantitation, and elucidation of underlying biosynthetic networks promises to advance mechanistic understanding of phenotypes and complement transcriptomics and proteomics[6–9].

However, the highly irregular and often unpredictable structures of metabolites pose a largely unmet challenge to their systematic chemical and biological annotation. High-performance liquid chromatography (HPLC)-HRMS analysis of a typical metabolome sample of plant or animal origin can reveal more than 100,000 molecular features (defined by a mass-to-charge ratio, m/z, and retention time, RT), representing a complex mixture of ions derived from known and unknown metabolites, adducts, naturally occurring isotopes and background[9–12]. Comparative analysis of samples representing different biological conditions or genetic backgrounds can identify molecular features that are significantly differential between conditions, akin to finding induced or repressed genes in transcriptomics. In the case of untargeted metabolomics, such comparative analyses provide the basis for prioritizing among the many detected unknowns for subsequent structure elucidation[13], which is often time- and resource-intensive.

Processing of metabolomics data involves three major steps – feature detection, comparative statistical analysis, and structural characterization – each of which comes with its own challenges addressed by free, open-source computational tools as summarized in Supplementary Table 1. Importantly, comparative metabolomics requires effective, multi-layered interaction with MS and MS fragmentation (MS/MS) data to enable culling and prioritization of differentially regulated molecular features, in particular when the analysis is discovery-oriented and focuses on the identification of unknowns. Exploring the metabolomes of C. elegans and other model systems, we recognized the need for an open-source analysis platform that can serve as a flexible and customizable hub integrating diverse existing tools.

We here introduce Metaboseek, a modular software platform that provides a comprehensive data analysis workflow, from feature detection to compound identification, specifically designed to facilitate untargeted metabolomics. Metaboseek incorporates popular metabolomics tools and makes them available in an intuitive, browser-based graphical user interface (Fig. 1, also see https://metaboseek.com/). We then leverage Metaboseek to investigate peroxisomal α-oxidation (pαo) in C. elegans, a conserved fatty acid degradation pathway that complements β-oxidation to enable the breakdown of β-branched fatty acids. Whereas β-oxidation takes place in both mitochondria and peroxisomes, α-oxidation occurs exclusively in peroxisomes, which are membrane-bound metabolic compartments that function coordinately with other organelles in lipid and bile acid metabolism[14]. In contrast to the well-studied peroxisomal β-oxidation pathway[15], the pαo pathway in C. elegans has not been investigated. Using Metaboseek for untargeted comparative metabolomics of a pαo mutant, we show that disruption of C. elegans pαo results in accumulation of several 100 metabolites, most of which have not previously been reported. The identified metabolites support homology-based annotation of the pαo pathway enzyme, HACL-1, and indicate a role for pαo in the processing of bacteria-derived cyclopropane fatty acids.

## Results

**The Metaboseek workflow.** In a typical metabolomics workflow, HPLC-HRMS and MS/MS data acquisition is followed by *Feature Detection* (a "feature" representing a specific m/z, RT pair) and *Feature Grouping* (aiming to recognize the same features across different samples, a non-trivial step, especially in the case of closely eluting isomers). Metaboseek integrates the *XCMS* package for feature detection and grouping, generating a feature table containing the data for all identified molecular features in each sample. Alternatively, feature tables can be imported from external tools, such as MZmine[16], MS-DIAL[17,18], or XCMSOnline[19]. Following feature detection and grouping, any subsequent statistical or data-dependent analysis generates new columns in the *Feature Table*, which serves as a customizable information hub that guides data analysis.

The *Data Explorer* section is at the heart of Metaboseek, providing the platform to further process, prioritize, sort, and visualize molecular features within the *Feature Table*. An extensive range of filters can be used in any user-defined configuration to enable intuitive prioritization of features, e.g., mean intensity threshold and fold change of a feature between different sample groups, at a specific significance level. Finding molecular features with significantly differential abundances in Metaboseek is assisted by built-in t-test and ANOVA, as well as group-wise fold-change metrics. For more advanced statistical analysis, feature tables can be exported directly in a MetaboAnalyst-compatible format[20]. Features of interest can be manually inspected in the interactive user interface, which includes a customizable *Data Viewer* for visual validation of molecular features, modules for *Molecular Formula Prediction* and isotope/adduct assignments, e.g., via CAMERA[21], and removal of background-derived features via *Peak Quality* analysis function, wherein features are scored based on fitting to an idealized peak shape. Following validation, Metaboseek can export annotated feature tables as inclusion lists to facilitate targeted MS/MS data acquisition. MS/MS spectra can be easily compared to each other using the *Keep and Compare* and *Find Patterns* functions (vide infra), or annotated using SIRIUS fragmentation trees and CSI:FingerID database matching[22,23].

The *Molecular Networking* tool in Metaboseek uses feature grouping information from *XCMS* to match MS/MS spectra with a corresponding molecular feature. In molecular networking, MS/MS spectra of different features are compared and ranked by similarity of fragmentation patterns[24,25]. Similarity scores are based on peak matching between pairs of averaged spectra, and subsequent calculation of the cosine score between relative peak intensities of the matched spectra, similar to the GNPS feature-based networking workflow[26]. The resulting networks, including spectra of individual features, can be easily viewed and evaluated in Metaboseek, which allows users to click nodes and compare MS/MS fragmentation patterns directly. Any information in the feature table, e.g., relative abundances of compounds across sample groups, comments, or statistical analyses, can be mapped onto the network view. Furthermore, the MS/MS networking parameters can be modified using the *Simplify Network* function, e.g., by adjusting similarity thresholds (edges), restricting the number of edges per node, or limiting the number of nodes per cluster (Supplementary Fig. 1).

The Metaboseek *Data Explorer* further incorporates an isotope tracing module, *Label Finder*, providing an integrated analysis option for stable isotope labeling experiments (Fig. 1). Like many popular open-source data analysis tools, Metaboseek is written in R, using the *shiny* R package for interactive data visualization, and thus can be run either on a server or locally on any computer. All data analysis steps are tracked so that settings can be archived and

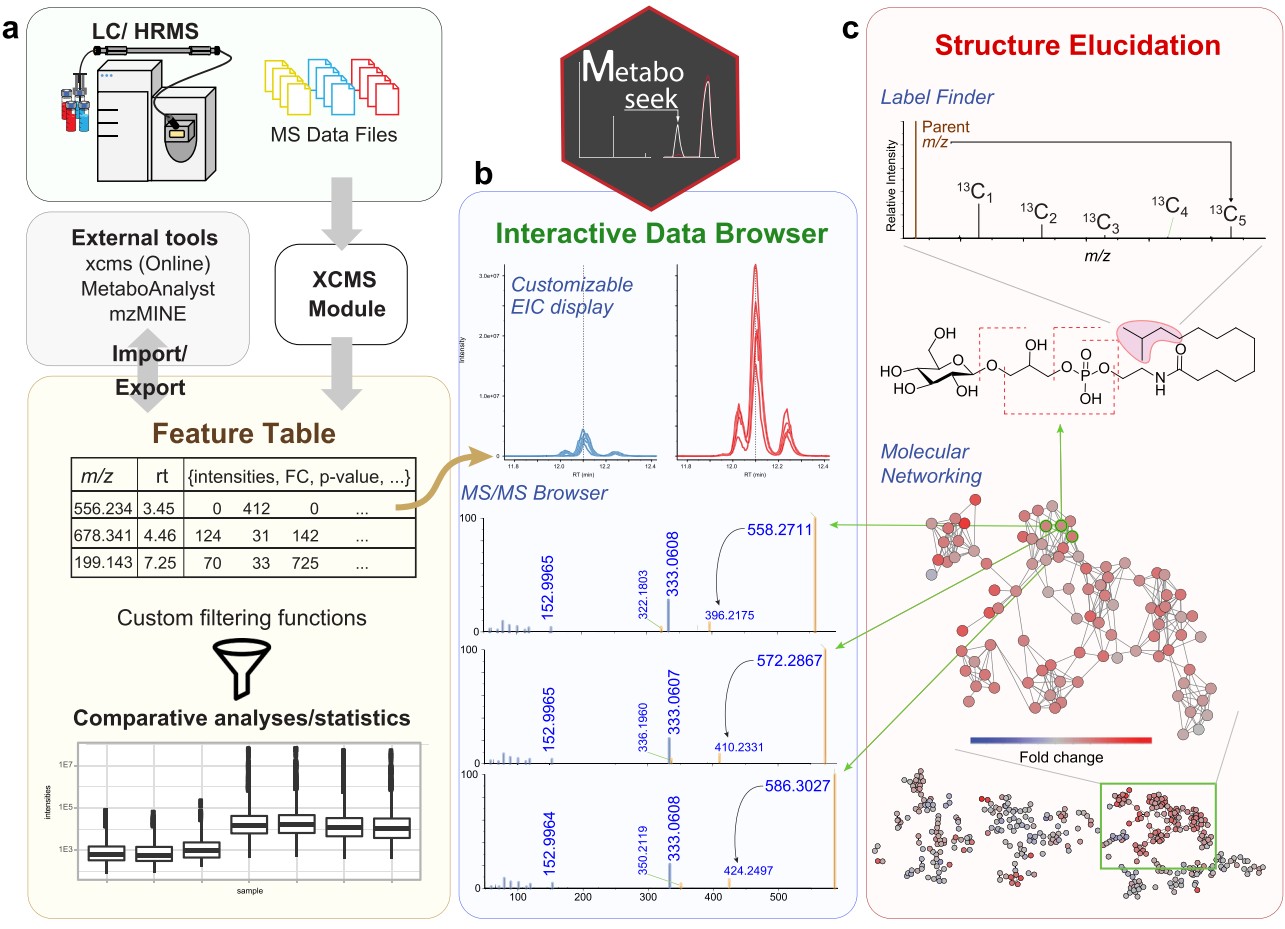

**Fig. 1 Comparative metabolomics with Metaboseek. a** Metaboseek includes an integrated XCMS module for feature detection and feature grouping (with CAMERA annotation) and accepts feature tables generated by other software. **b** Features can be annotated and prioritized using extensive filtering options and integrated statistics tools. Raw data for each molecular feature can be browsed rapidly, including associated EICs, MS1, and MS/MS spectra. **c** The data browser interacts with a suite of structure elucidation tools, e.g., SIRIUS-based molecular formula and structure prediction, the *Label Finder* to identify isotope-labeled compounds, and the MS/MS pattern finder to identify MS features with characteristic fragmentation patterns.

reproduced. Installation files and extensive documentation are available online at https://metaboseek.com/.

**Citronellic acid as a probe for α-oxidation.** A putative *C. elegans* α-oxidation pathway was proposed more than two decades ago, based on sequence similarity to characterized rat and human enzymes (Fig. 2a)[27]. In humans, the principal function of α-oxidation is presumed to be the enzymatic digestion of β-branched fatty acids (**1**), such as (*R*)-citronellic acid (CA, **6**) (Fig. 2d), which cannot be processed by β-oxidation[28]. In the first step of the human α-oxidation pathway, fatty acyl-coenzyme A (CoA) derivatives (**2**) are *syn*-hydroxylated at the α-position by phytanoyl-CoA dioxygenase, PHYH, an iron-coordinating enzyme that is 52% and 59% identical to the uncharacterized *C. elegans* proteins, ZK550.5 and ZK550.6, respectively (Fig. 2a)[29]. Genetic mutations in *PHYH* cause Refsum disease, which is characterized by toxic accumulation of branched fatty acids in the blood and nervous system[30]. The next step is catalyzed by 2-hydroxyacyl-CoA lyase, HACL1, which binds thiamine pyrophosphate (TPP) as a cofactor and cleaves a C-C bond in the α-hydroxy, β-methylacyl-CoA (**3**) to produce formyl-CoA and an α-methyl fatty aldehyde (**4**). The *C. elegans* gene *B0334.3*, herein referred to as *hacl-1*, encodes an enzyme 49% identical to human HACL1, including high homology in the TPP-binding domain (Supplementary Fig. 2). Finally, the aldehyde (**4**) is oxidized in an NAD$^+$-dependent reaction to the corresponding

α-methyl fatty acid (**5**), which now is a suitable substrate for further processing via β-oxidation (Fig. 2a).

To probe pαo in *C. elegans*, we compared metabolism of supplemented CA (**6**) in WT animals and *hacl-1(tm6725)* mutants (Fig. 2b), which harbor a deletion predicted to disrupt the splice acceptor site of the largest exon (Fig. 2c). *hacl-1(tm6725)* mutants develop normally and exhibit no overt abnormalities under laboratory conditions. If *hacl-1* were required for α-oxidation, supplementation with CA should result in accumulation of an α-hydroxyl CA derivative (**7**), whose MS/MS spectrum should show a characteristic neutral loss of formic acid (Fig. 2d)[31]. The custom filtering options and extracted ion chromatogram (EIC) display in Metaboseek facilitated screening molecular ions at the expected *m/z* of **7** (*m/z* 185.1183, $C_{10}H_{17}O_3^-$) that produce neutral loss of formic acid during MS/MS and were strongly enriched in *hacl-1* samples (Fig. 2e). Of six other CA-derived features detected at *m/z* 185.1183, none were enriched in *hacl-1* relative to WT, nor did these features exhibit neutral loss of formic acid in MS/MS, suggesting these features are derived from hydroxylation of CA via other metabolic pathways (Supplementary Fig. 3).

Additional CA-dependent features enriched in *hacl-1* relative to CA-supplemented WT animals were uncovered using a combination of fold change, intensity, and sample group-dependent filters, which, after removal of adducts, revealed 32 CA-derived metabolites (Supplementary Fig. 3). These differential

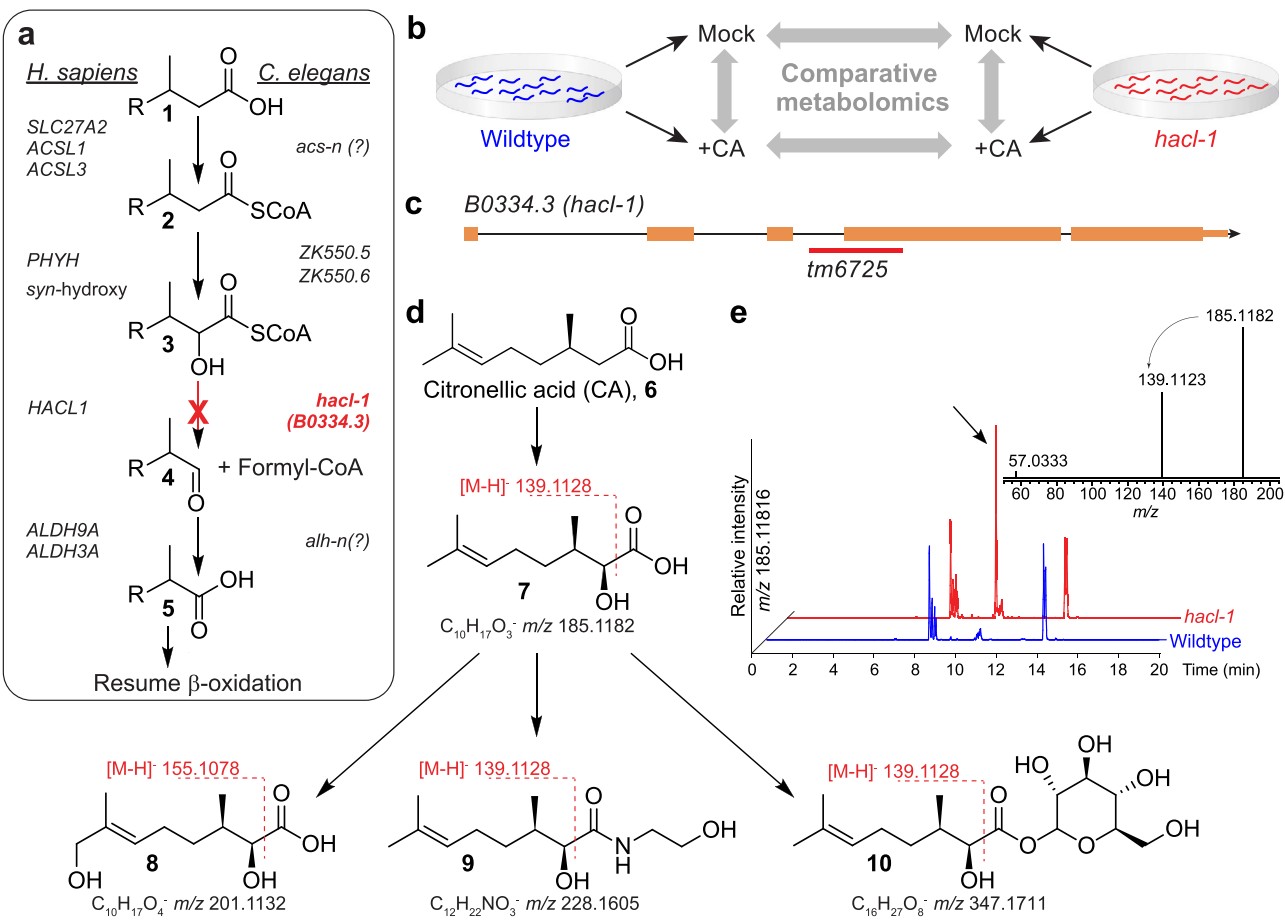

**Fig. 2 Comparative metabolomics of CA-fed *hacl-1* mutants. a** Conservation of peroxisomal α-oxidation in *C. elegans* and humans. **b** CA-feeding experiment. *C. elegans* larvae were supplemented with CA, (**6**) followed by comparative analysis with Metaboseek. **c** *hacl-1(tm6725)* mutants harbor a genomic deletion (red line) spanning the indicated exon splice junction, orange rectangles represent exons, black lines represent introns. **d** Representative shunt metabolites accumulating in *hacl-1* following CA supplementation. The majority of shunt metabolites fragment between the carbonyl- and α-carbon during MS/MS, which is characteristic of α-hydroxy fatty acids. **e** HPLC-MS (negative ion) EIC for *hacl-1*-enriched feature with *m/z* 185.1182, corresponding to **7** (arrow). Its MS/MS spectrum reveals a product ion at *m/z* 139.1123, corresponding to neutral loss of formic acid.

features included multiply oxygenated CA derivatives, which could be ostensibly derived from ω-oxidation following stalled α-oxidation[32]. For example, we detected several features at *m/z* 201.1132 that elute over a wide RT range, representing dihydroxylated CA derivatives, two of which were *hacl-1*-enriched (such as **8**, see also Supplementary Fig. 3). Additional *hacl-1*-dependent CA derivatives included putative ethanolamides (**9**), glycosides (**10**), and an *N*-acyl glycerophosphoethanolamine conjugate (Supplementary Fig. 3). For most of these compounds, MS/MS fragmentation between the α- and carbonyl carbons suggested α-hydroxylation of the citronellyl moiety (Fig. 2d). Taken together, this supplementation study revealed a set of *hacl-1*-dependent shunt metabolites of CA, consistent with the proposed function of HACL-1 as a 2-hydroxyacyl-CoA lyase.

**Endogenous $C_{11}$ fatty acids enriched in *hacl-1* larvae.** We next investigated the impact of *hacl-1* inactivation on endogenous metabolites, using molecular networking of MS/MS spectra acquired as part of in-depth untargeted HPLC-HRMS analysis of *hacl-1* mutants and WT *C. elegans*. To assess the role of *hacl-1* in *C. elegans* metabolism, we initially focused on starved animals at the first larval stage (L1), a condition that allows the study of *C. elegans* metabolism in the absence of bacterial food[33]. Comparative analysis of negative ion MS data for L1 larvae revealed a

small set of features strongly enriched in *hacl-1* relative to WT, several of which clustered together in the MS/MS network (Fig. 3a). Stringent fold change (10-fold), intensity (top 1.25% of detected features), and unadjusted significance ($p < 0.05$) thresholds yielded 57 molecular features that were highly enriched in *hacl-1* larvae. Following *CAMERA* isotope/adduct assignment and manual curation, we detected 14 *hacl-1*-dependent metabolites (Fig. 3b). Intriguingly, the majority of these compounds appeared to represent $C_{11}$ fatty acids, based on their ionization properties and MS/MS spectra. Furthermore, MS/MS spectra of the most abundant *hacl-1*-enriched metabolites featured a product ion with *m/z* 72.993 corresponding to glyoxylate ($C_2HO_3^-$), which was not observed in any CA-derived metabolites (Fig. 3c).

We selected one highly abundant *hacl-1*-enriched metabolite ($C_{11}H_{19}O_4^-$) for isolation by preparative HPLC followed by structure elucidation via 2D NMR spectroscopy (Supplementary Fig. 4 and Supplementary Table 2), which revealed an unusual β-methyl substituted, eleven-carbon fatty acid, named bemeth#3 (**11**, Fig. 3c). α-Hydroxylation and the position of the double bond in bemeth#3 are consistent with the strong glyoxylate product ion in its MS/MS spectrum. Moreover, the structure of bemeth#3 suggested that other glyoxylate ion-producing metabolites that accumulate in *hacl-1* mutants also represent derivatives of α-hydroxylated β-methyl-4-decenoic acid (e.g., **12**, Fig. 3c). These assignments were further supported via synthesis of an authentic

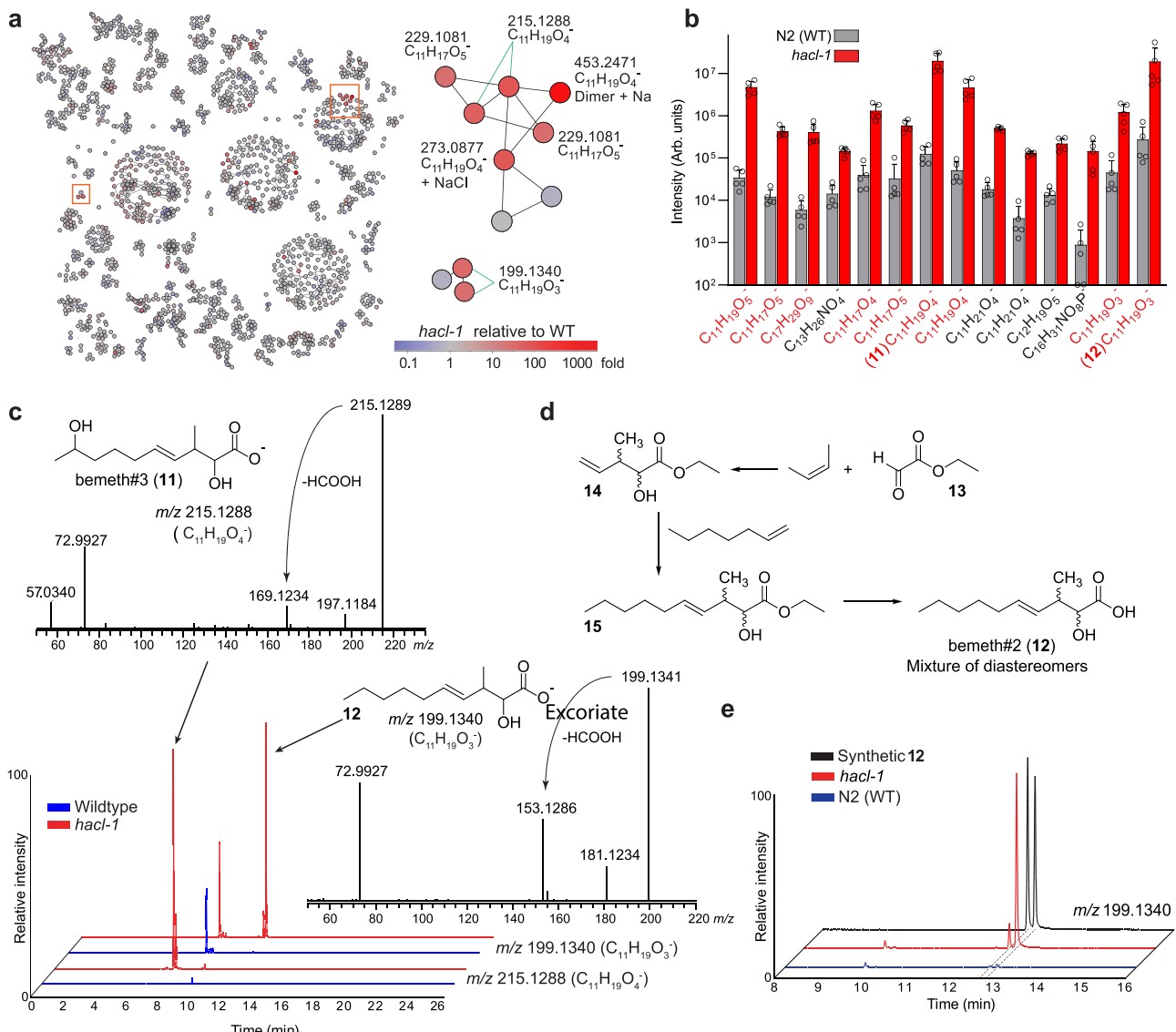

**Fig. 3 Endogenous metabolites accumulating in *hacl-1* mutant larvae. a** MS/MS network highlighting differential abundance in *hacl-1* mutants relative to WT. Subnetworks of interest (orange boxes) are shown enlarged. **b** Quantification of endogenous metabolites ten-fold or more enriched in *hacl-1* larvae relative to WT at $p < 0.05$ as calculated by unpaired two-sided *t*-test, unadjusted for number of comparisons. Metabolites in red produce an MS/MS product ion with *m/z* 72.993. Data represent five independent experiments and bars means ± standard deviation. **c** Representative HPLC-MS (negative ion) EICs and MS/MS spectra for most abundant differential metabolites in **b** bemeth#3 (**11**) and bemeth#2 (**12**). **d** Overview of synthetic scheme to afford bemeth#2 (see Methods for details). **e** Comparison of HPLC-MS (negative ion) EICs for synthetic diastereomers of bemeth#2 (**12**) and the corresponding metabolites in *exo*-metabolome extracts from WT and *hacl-1* larvae. Source data are provided as a Source Data file.

sample of the two diastereomers of **12** (Fig. 3d, Supplementary Methods, and Supplementary Table 3), whose MS/MS spectra and retention times matched those of the corresponding *hacl-1*-enriched metabolites (Fig. 3e). Other metabolites enriched in *hacl-1* include less abundant isomers of **11** and **12** with identical MS/MS fragmentation, as well as derivatives that appear to have undergone additional oxidation, including putative dicarboxylic acids, such as $C_{11}H_{17}O_5^-$ (Supplementary Fig. 5). Taken together, analysis of *hacl-1* larvae revealed an unusual family of $C_{11}$ fatty acids based on the β-methyl-decenoic acid scaffold, which has not been previously reported from animals. The presence of an α-hydroxyl group in the identified $C_{11}$ acids **11** and **12** suggests that they represent plausible substrates of HACL-1 and therefore accumulate in *hacl-1* mutants. The more highly oxygenated derivatives, such as $C_{11}H_{17}O_5^-$, could result from ω-oxidation of **12** as part of a shunt pathway, similar to the role of ω-oxidation in human fatty acid metabolism[32].

**Comparative metabolomics of *hacl-1* adults.** Next, we employed Metaboseek for comparative metabolomics of adult-stage *hacl-1* mutant and WT animals. Conditioned culture medium (*exo*-metabolome) and worm bodies (*endo*-metabolome) were harvested separately, extracted, and analyzed by HPLC-HRMS/MS in positive and negative ionization modes, yielding more than 700,000 molecular features combined, of which ~150,000 remained after blank subtraction and *Peak Quality* thresholding (see Methods and Supplementary Data 1). Like *hacl-1* mutant larvae, *hacl-1* adults accumulate the β-branched $C_{11}$ acid **11** and related metabolites (Supplementary Fig. 6). However, in contrast to L1 larvae, these $C_{11}$ acid derivatives were not the most differential metabolites in *hacl-1* adults (Supplementary Fig. 6). Untargeted analysis using intensity (top 12% of remaining features), unadjusted significance ($p < 0.05$), and fold-change (5-fold) filters uncovered >1000 features that were enriched in the *hacl-1 exo*-metabolome, which we explored by MS/MS networking.

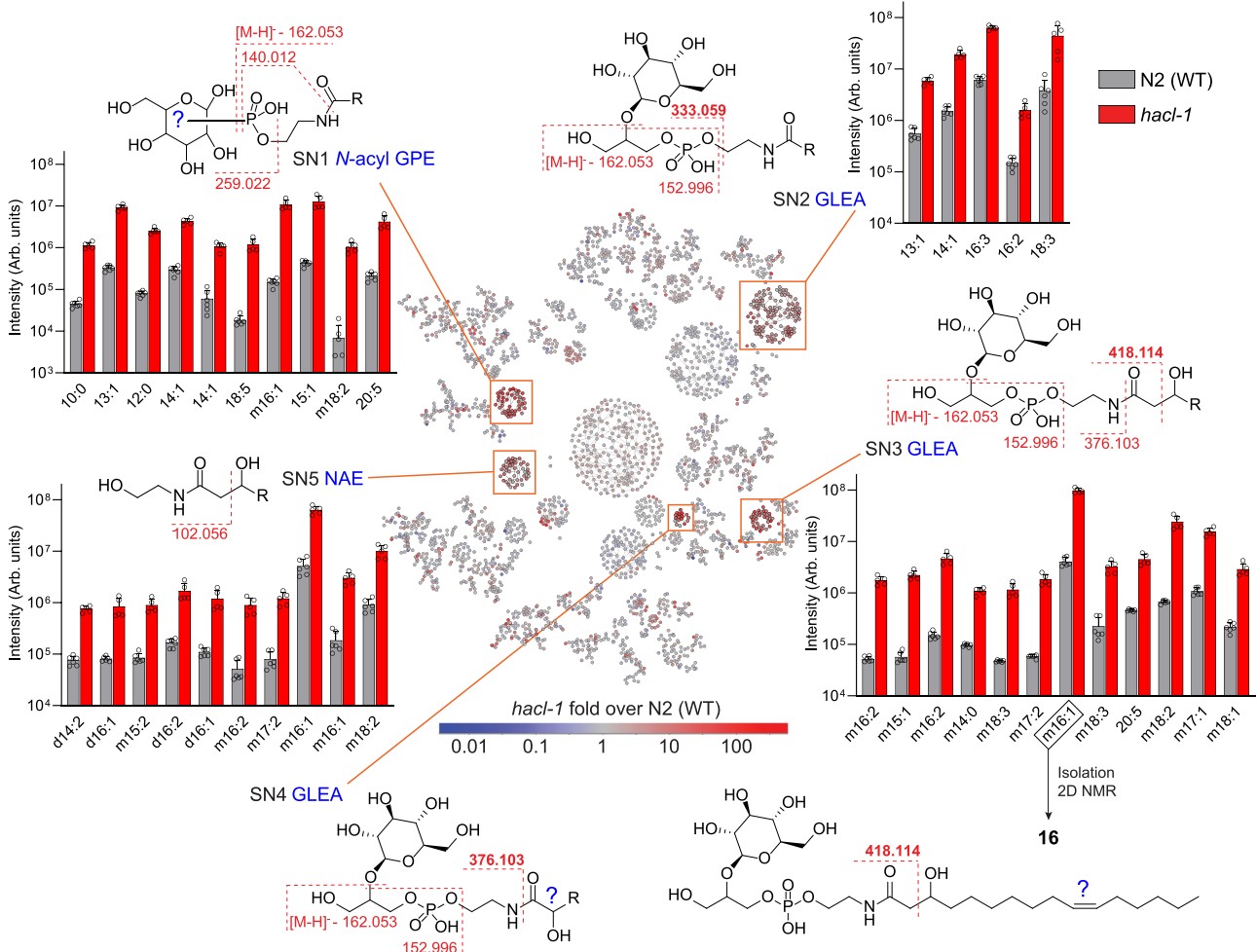

**Fig. 4 MS/MS network comparing *exo*-metabolomes of *hacl-1* and WT adults.** Proposed structures and major fragmentation reactions are shown for five subnetworks (SN1 – SN5, orange boxes). Example compounds in the bar graphs are at least 10-fold enriched in *hacl-1* mutants relative to WT and satisfy mean intensity criteria (10$^6$ for SN1, SN2, SN3; 5 ×10$^5$ for SN5). R represents an acyl group with N carbons and n degrees of unsaturation (N:n), preceded by *m* or *d* for *m*ono- or *d*i-oxygenated. The structure of the most abundant metabolite in SN3, GLEA-m16:1 (**16**), was characterized via 2D NMR spectroscopy. For compounds from SN4, see Supplementary Fig. 10. Data represent six (WT) or five (*hacl-1*) samples from three biologically independent experiments and bars means ± standard deviation. Source data are provided as a Source Data file.

The majority of differential features clustered in five major subnetworks (SN1-SN5, Fig. 4). Inspection of SN1 revealed several homologous series of features, related by the mass difference of a methylene (Δ*m/z* 14.0156). Use of the "*Keep and Compare*" functionality in the *MS/MS Browser* facilitated simultaneous display of multiple MS/MS spectra and automatically highlights fragments shared between spectra, e.g., conserved product ions that correspond to a phosphoethanolamine moiety, a phosphorylated hexose, and ions indicating a phosphate group (Supplementary Fig. 7). Complementary analysis of MS/MS fragmentation patterns in positive ionization mode further supported that SN1 represents *N*-acyl glycophosphoethanolamides ("*N*-acyl GPEs"), including saturated (N:0), singly unsaturated (N:1), polyunsaturated (N:n), and mono-oxygenated (mN:n) acyl moieties ranging from C$_8$-C$_{20}$. In many cases, several isobaric features were detected, e.g., four distinct isomers of *N*-acyl GPE-14:2 (Supplementary Fig. 8). Enrichment trends were similar across the *endo*- and the *exo*-metabolomes, but *N*-acyl GPE were 10–100-fold more abundant in the *exo*-metabolome (Supplementary Fig. 8).

**Finding MS/MS patterns with Metaboseek.** Analysis of SN2 indicated that it represents *N*-acyl glycoglycerophosphoethanolamides ("GLEA", Fig. 4). This compound family had

been previously described in the context of ethanol-dependent de novo fatty acid biosynthesis in starved *C. elegans* larvae[34]. MS/MS spectra of GLEA exhibit a characteristic product ion at *m/z* 333.0592, in addition to phosphoglycerol and several glucose-derived fragments, which were used to define a "*Pattern*" in Metaboseek. The *Find Patterns* function in Metaboseek searches all MS/MS spectra for user-defined fragmentation patterns, including neutral losses; hits are matched and recorded in new interactive columns in the feature table, which enabled rapid identification of more than 100 GLEA-like molecular features distributed across SN2 and two additional subnetworks, SN3 and SN4 (Fig. 4).

Each of the three GLEA clusters revealed slightly different MS/MS fragmentation, providing important structural clues. GLEA in SN3 produced additional product ions with *m/z* 376.1016 and 418.1123, suggesting that these metabolites are β-hydroxylated, which then results in fragmentation between the α and β carbons (Fig. 4). Larger cultures of *hacl-1* were grown and extracted to isolate the most abundant compound from SN3, GLEA-m16:1, one of the most intense and differential features in the entire *exo*-metabolome. 2D NMR spectroscopic analysis established this compound as a 2-*O*-(β-glucosyl)-glycero-1-phosphoethanolamide of β-hydroxylated hexadecenoic acid (**16**, Supplementary Table 4).

β-hydroxylation of the fatty acyl moiety in this compound is consistent with observed fragmentation between the α and β carbons of the fatty acyl group and suggests that other metabolites in SN3 also represent GLEA of β-hydroxylated fatty acids (Fig. 4). Upregulation of β-hydroxylated lipid derivatives suggested that mitochondrial or peroxisomal β-oxidation may be perturbed in *hacl-1* mutants. However, production of ascaroside pheromones, which relies on peroxisomal β-oxidation, was largely unchanged in *hacl-1* mutants compared to WT (Supplementary Fig. 9), suggesting that *hacl-1* inactivation may interact with mitochondrial β-oxidation.

GLEA in SN4 did not undergo fragmentation across the α and β carbons, but instead produced an intense product ion with *m/z* 376.1016, corresponding to fragmentation across the amide bond. GLEA in SN4 were much less abundant and eluted later than isobaric metabolites in SN3. We hypothesize that SN4 represents GLEA bearing α-hydroxy acyl substituents; however, their low abundance precluded NMR spectroscopic characterization (Supplementary Fig. 10). Lastly, analysis of SN5 revealed a large family of *hacl-1*-enriched *N*-acyl ethanolamides (NAEs) (Fig. 4). All NAE in SN5 produced the product ion with *m/z* 102.056, corresponding to cleavage between the α- and β-carbon of the acyl group, suggesting β-hydroxylation in analogy to SN3.

**Stable isotope tracing with Metaboseek.** Comparing the series of *N*-acyl-GPEs, GLEA, and NAEs enriched in *hacl-1* worms, we noted that derivatives of mono-unsaturated $C_{13}$- and $C_{15}$-fatty acids were among the most abundant *hacl-1*-enriched compounds, even though the corresponding free fatty acids are not particularly abundant in *C. elegans*[35]. Generally, odd-chain fatty acids in *C. elegans* are derived primarily either from iso-branched chain fatty acid (BCFA) biosynthesis, which employs leucine-derived isovaleryl-CoA as a starter unit[36], or, alternatively, from metabolism of diet-derived cyclopropane fatty acids, which are abundantly produced by the bacterial diet, *E. coli* OP50[37].

We first tested whether the major *hacl-1*-enriched mono-unsaturated $C_{13}$ and $C_{15}$ lipids are derived from BCFA metabolism. For this purpose, we grew worms supplemented with $^{13}C_6$-labeled leucine, which we reasoned should result in $^{13}C_5$-enrichment of BCFAs and any derived *N*-acyl GPE, GLEA, and NAE (Fig. 5a). The *Label Finder* tool in Metaboseek facilitated profiling $^{13}C_5$- and $^{13}C_6$-enrichment for discovery of BCFA- and Leu-derived metabolites. This analysis revealed several hundred isotope-enriched features, including iso-branched fatty acids and derivatives thereof, which were visually validated using the *Mass Shifts* feature in Metaboseek to display EICs corresponding to incorporation of $^{13}C_5$ (Δ*m/z*, 5.0167, Fig. 5b). However, the most abundant *hacl-1*-enriched compounds harboring 13:1 and 15:1 acyl groups showed no evidence for label incorporation, indicating that these unsaturated odd chain lipids do not originate from BCFA metabolism (Supplementary Fig. 11).

**Cyclopropane fatty acid derivatives accumulate in *hacl-1*.** We then asked whether the monounsaturated $C_{13}$ and $C_{15}$ fatty acyl derivatives accumulating in *hacl-1* mutants are derived from bacterial cyclopropane fatty acids. In the case of *E. coli* OP50, $C_{17}$ and $C_{19}$ cyclopropane lipids can account for nearly half of all lipid species and thus comprise a substantial portion of *C. elegans* lipid intake[38]. To test whether the monounsaturated $C_{13}$ and $C_{15}$ lipids enriched in *hacl-1* are derived from cyclopropane fatty acids, we compared the metabolomes of worms fed either OP50 or JW1653-1 bacteria, a cyclopropane-deficient *E. coli* strain[39] (Fig. 5c). First, we confirmed via 2D NMR spectroscopy that JW1653-1 does not produce cyclopropane lipids and that worms

fed JW1653-1 bacteria do not produce cyclopropane lipids (Supplementary Fig. 12). Next, we compared the metabolomes of animals grown on OP50 or JW1653-1 via HPLC-HRMS, which revealed that production of the most abundant *hacl-1*-enriched *N*-acyl GPEs was abolished in JW1653-1-fed worms (Fig. 5d). Additional *N*-acyl GPEs enriched in *hacl-1* mutant were also found to be dependent on bacterial cyclopropane fatty acid biosynthesis, including multiple hydroxylated *N*-acyl GPE species (Fig. 5e). Untargeted comparative analysis of OP50- and JW1653-1-fed worms revealed a large number of other cyclopropane-containing metabolites, including GLEA, as well as putative oxidized fatty acids and fatty acyl glycosides, many of which also accumulate in *hacl-1* worms (Fig. 5f and Supplementary Fig. 13). Taken together, our results suggest that diet-derived $C_{17}$ or $C_{19}$ cyclopropane fatty acids are initially chain shortened via β-oxidation to yield shorter chained derivatives that become substrates for pαo. If pαo is blocked, as in the case of *hacl-1* inactivation, β-oxidation intermediates are shunted towards production of, e.g., *N*-acyl GPE, GLEA, and other lipids (Fig. 5g).

## Discussion

We here demonstrated the use of Metaboseek for a multi-layered comparative metabolomics study of a conserved fatty acid metabolism pathway, pαo, in *C. elegans*. By probing metabolism of WT and *hacl-1* mutants with a pαo test substrate, CA, we confirmed the predicted enzymatic function of HACL-1 as a 2-hydroxyacyl-CoA lyase. Subsequent untargeted comparison revealed pervasive changes in lipid metabolism in *hacl-1* mutants, including accumulation of an unusual family of α-hydroxylated β-branched $C_{11}$ acids. Their abundant production and life stage-specific regulation suggests that β-branched $C_{11}$ acids – perhaps a precursor or downstream metabolite of **11** and **12** – may serve specific functions in *C. elegans*. In addition to the $C_{11}$ acids, *hacl-1* mutants accumulate an unexpected diversity of modular lipids derived from the intersection of multiple branches of fatty acid metabolism with NAE biosynthesis. Particularly abundant among lipids accumulating in *hacl-1* are derivatives of cyclopropyl fatty acids, suggesting that pαo participates in cyclopropyl metabolism. Mechanisms for the breakdown of cyclopropyl fatty acids have remained largely unknown, though cyclopropyl lipids have previously been shown to affect recovery from larval diapause (dauer)[40]. Cyclopropyl lipids are also present in the human diet, most prominently in cheese and dairy originating from animals fed fermented grains; however, whether α-oxidation plays a role in the mammalian metabolism of cyclopropyl lipids remains to be determined[41,42].

Comparative analysis with Metaboseek revealed a large number of additional differences between the metabolomes of WT and *hacl-1* animals, of which many represent previously undescribed metabolites (Supplementary Data 2). For all newly annotated metabolites, this table includes retention time, *m/z*, putative molecular formulae and compound class assignments, fold-change, as well as isotopic enrichment data, which will facilitate follow-up studies in conjunction with the deposited MS raw data[43–45]. As we here demonstrated, in-depth evaluation of MS and MS/MS raw data is key to discovery-oriented workflows. Online resources such as GNPS and MassBank provide access to vast amounts of MS data, highlighting the need for versatile tools that facilitate raw data analysis for metabolomics[46–48]. For this purpose, Metaboseek combines more than 60 different modules which tool developers can use as building blocks for specialized data analysis apps with minimal effort. The seamless integration of intuitive data filters and a range of analysis tools facilitates metabolite annotation up to confidence Level 3 for many detected features, enabling tentative structure or compound class

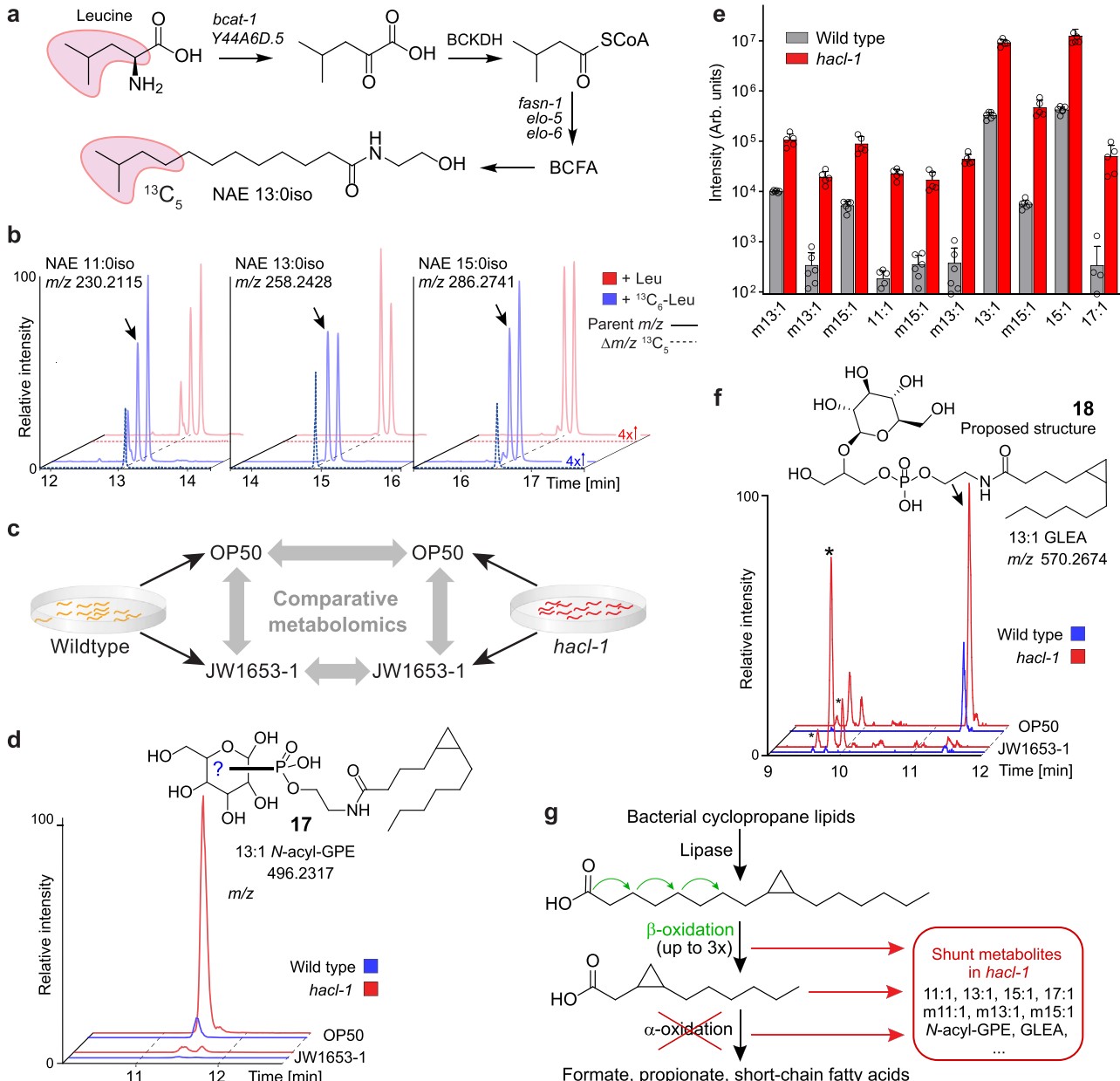

**Fig. 5 Cyclopropane-containing glycolipids are enriched in *hacl-1* mutants. a** Leucine metabolism feeds into branched-chain fatty acid (BCFA) biosynthesis in *C. elegans*. **b** Pairwise analysis of Leu or $^{13}C_6$-Leu-supplemented worms using the *Label Finder* revealed $^{13}C_5$-enriched NAEs derived from BCFAs. Shown are representative HPLC-MS (ESI+) EICs and dotted lines represent incorporation of $^{13}C_5$ ($\Delta m/z$, 5.0167), as visualized using Metaboseek *Mass Shifts*. **c** Study design for comparative metabolomics of WT and *hacl-1* worms fed either *E. coli* OP50 or cyclopropane fatty acid-deficient *E. coli* JW1653-1. **d** Representative HPLC-MS (ESI-) EIC for 496.2317, corresponding to cyclopropane-containing *N*-acyl GPE 13:1 (**17**). Shown structure was proposed based on MS/MS fragmentation and absence in JW1653-1. **e** Quantification of *N*-acyl GPEs that were absent from worms fed JW1653-1. Data represent six (WT) or five (*hacl-1*) samples from three biologically independent experiments and bars means ± SD. **f** Representative HPLC-MS (ESI-) EIC for *m/z* 570.2674, corresponding to cyclopropane-containing GLEA 13:1 (**18**). Shown structure was proposed based on MS/MS fragmentation and absence in JW1653-1. Asterisks (*) mark isobaric *hacl-1*-enriched features that are not impacted by JW1653-1 diet. **g** Proposed metabolism of bacterial 17:1 (or 19:1) cyclopropane lipids. Three (or four) rounds of β-oxidation would produce an 11:1 cyclopropane fatty acid unsuitable for further β-oxidation that could be a substrate for α- or ω-oxidation. Source data are provided as a Source Data file.

assignments[49–51]. In parallel, facile statistical analysis of metabolite variation across multiple genotypes and/or environmental conditions enables developing functional and biosynthetic hypotheses. Further, the output from Metaboseek facilitates intersecting metabolomics with transcriptomics, proteomics, or genomics, toward a systems-level understanding of biosynthetic networks and metabolite functions. Metaboseek thus provides a flexible and expandable open-source platform to accelerate

chemical and biological annotation of metabolites, including the large space of yet unidentified biogenic small molecules.

Our comparative metabolomics analysis of α-oxidation shows that even a primary metabolic pathway in an otherwise well-studied model system can reveal a large number of previously uncharacterized compounds, as well as unexpected connections to other pathways, e.g., β-oxidation or cyclopropane fatty acid metabolism. Like much of conserved primary metabolism,

α-oxidation was initially characterized more than 50 years ago[52,53]. It seems likely that re-analysis of primary metabolic pathways using state-of-the-art HRMS and data analysis tools will synergize with transcriptomic and proteomic studies to harness the potential of metabolomics as the 'omics discipline that most closely reflects phenotype[7–9].

## Methods

***C. elegans* strains.** Unless otherwise indicated, worms were maintained at 20 °C on Nematode Growth Medium (NGM) 6 cm diameter petri dish plates seeded with *E. coli* OP50 obtained from the *Caenorhabditis* Genetics Center (CGC). For experiments with cyclopropane deficient bacteria, worms were grown on NGM 6 cm plates seeded with *E. coli* JW1653-1, a kind gift from the Walhout Lab (University of Massachusetts Medical School, Worcester, MA). The following *C. elegans* strains were used for comparative metabolomics: Bristol N2 ("wildtype") obtained from the CGC, and *B0334.3(tm6725)* obtained from the National Bioresource Project, Tokyo, Japan[54], referred to as *hacl-1*, strain designation FCS7. The FCS7 strain was the result of backcrossing *tm6725* with *sqt-1* (FCS6) for seven generations. FCS6 was iteratively backcrossed with Bristol N2 for a total of six generations. After the final backcross, FCS7 hermaphrodites were singled and allowed to self, non-rollers were picked, and the genotype was confirmed by PCR and Sanger sequencing (Cornell University Institute of Biotechnology). The primers used for PCR (Integrated DNA Technologies) included forward: GAAGTAGGAATGGCAGCACAAG, reverse: GGAACTGCTGAACTTGTGTAGCTC, and an additional primer in the deleted genomic interval: CTGCTGGCCTGTAGTCTGTATTG.

**Test substrate feeding experiments.** Alkaline bleach treatment of mixed-stage animals yielded a sterile egg suspension, which was rocked overnight in 3 mL M9 solution to yield synchronized, starved L1 larvae[55]. Approximately 100,000 synchronized L1 larvae were added to 125 mL Erlenmeyer flasks containing 10 mL M9 media and 300 μM citronellic acid (Sigma-Aldrich 303429), phytanic acid (Sigma-Aldrich P4060), retinoic acid (Sigma-Aldrich R2625), geranic acid (Aldrich 427764), or an equivalent volume of methanol only (vehicle control) and were incubated at 20 °C with shaking at 180 RPM for 24 h. Cultures were transferred to 15 mL conical tubes and centrifuged (500×*g*, 22 °C, 1 min), and the resulting supernatant (*exo*-metabolome) was transferred to a fresh conical tube and snap frozen. Remaining L1 pellet was washed three times with M9 before snap freezing in liquid nitrogen.

***C. elegans* liquid cultures.** For the analysis of staged gravid adults, ~75,000 synchronized L1 larvae obtained from alkaline bleach treatment were added to 125 mL Erlenmeyer flasks containing 25 mL S-complete medium and kanamycin at 35 μg/mL to prevent contamination. Worms were fed with 50x concentrated *E. coli* OP50 or *E. coli* JW1653-1 and incubated at 20 °C with shaking at 180 RPM for 66–70 h, at which time the population was predominantly gravid adults, determined by microscopic inspection. Control samples to account for bacterial matrix were prepared with the same amount of *E. coli* OP50 or JW1653-1 under identical conditions. Liquid cultures were transferred to 50 mL conical tubes and centrifuged (500 × g, 22 °C, 1 min), and the top 20 mL of the resulting supernatant (*exo*-metabolome) was transferred to a fresh conical tube and snap frozen. Remaining worm pellet was transferred to a 15 mL conical tube, centrifuged (500 × *g*, 22 °C, 1 minute), and washed three times with M9 before snap freezing in liquid nitrogen.

**$^{13}C_6$-Leu isotope tracing experiment.** Approximately 60,000 synchronized N2 (WT) L1 larvae obtained from alkaline bleach treatment were added to 125 mL Erlenmeyer flasks containing 20 mL S-Complete medium. Worms were fed with 60 mg freeze-dried OP50 powder (InVivoBiosystems, formerly NemaMetrix Inc., OP-50-31772) and supplemented with leucine (Sigma Aldrich L8000) or $^{13}C_6$-leucine (Cambridge Isotope Laboratories CLM-2262-H-PK) at a final concentration of 2 mM. Worms were incubated at 20 °C with shaking at 180 RPM for 66–70 hrs, at which time the population was a mixture of young and gravid adults, determined by microscopic inspection. Liquid cultures were centrifuged (500 × *g*, 22 °C, 1 min), and the resulting supernatant was snap frozen. Worm pellet was washed three times with M9 before snap freezing in liquid nitrogen.

**Sample preparation for HPLC-MS.** *Exo*-metabolome (conditioned media) samples were lyophilized ~48 hrs using a VirTis BenchTop 4 K Freeze Dryer. Dried material was directly extracted in 10 mL methanol in 20 mL glass vials stirred overnight. Vials were centrifuged at 2750 × *g* for 5 min in an Eppendorf 5702 Centrifuge using rotor F-35-30-17. The resulting supernatant was transferred to a clean 20 mL glass vial and concentrated to dryness in an SC250EXP Speedvac Concentrator coupled to an RVT5105 Refrigerated Vapor Trap (Thermo Scientific). The resulting powder was suspended in methanol and analyzed directly by HPLC-MS, as described below. *Endo*-metabolome (nematode bodies) were lyophilized for 18-24 hrs using a VirTis BenchTop 4 K Freeze Dryer. Dried pellets were transferred to 1.5 mL microfuge tubes and disrupted in a Spex 1600 MiniG tissue grinder after the addition of two stainless steel grinding balls to each sample.

Microfuge tubes were placed in a Cryoblock (Model 1660) cooled in liquid nitrogen, and samples were disrupted at 1100 RPM for 60 s. This process was repeated two additional rounds for a total of three disruptions. Pellets were transferred to 8 mL glass vials in 5 mL methanol and stirred overnight. Subsequent steps for concentration and resuspension were followed as described for the *exo*-metablome.

**Mass spectrometry.** Liquid chromatography was performed on a Vanquish HPLC system controlled by Chromeleon Software (ThermoFisher Scientific) and coupled to an Orbitrap Q-Exactive High Field mass spectrometer controlled by Xcalibur software (ThermoFisher Scientific). Methanolic extracts prepared as described above were separated on a Thermo Hypersil Gold C18 column (150 mm × 2.1 mm, particle size 1.9 μM; 25002-152130) maintained at 40 °C with a flow rate of 0.5 mL/min. Solvent A: 0.1% formic acid (Fisher Chemical Optima LC/MS grade; A11750) in water (Fisher Chemical Optima LC/MS grade; W6-4); solvent B: 0.1% formic acid in acetonitrile (Fisher Chemical Optima LC/MS grade; A955-4). A/B gradient started at 1% B for 3 min after injection and increased linearly to 98% B at 20 min, followed by 5 min at 98% B, then back to 1% B over .1 min and finally held at 1% B for the remaining 2.9 min to re-equilibrate the column (28 min total method time). Mass spectrometer parameters: spray voltage, −3.0 kV/+3.5 kV; capillary temperature 380 °C; probe heater temperature 400 °C; sheath, auxiliary, and sweep gas, 60, 20, and 2 AU, respectively; S-Lens RF level, 50; resolution, 120,000 at *m/z* 200; AGC target, 3E6. Each sample was analyzed in negative (ESI−) and positive (ESI+) electrospray ionization modes with *m/z* range 100–1000. Parameters for MS/MS (dd-MS2): MS1 resolution, 60,000; AGC Target, 1E6. MS2 resolution, 30,000; AGC Target, 2E5. Maximum injection time, 60 msec; Isolation window, 1.0 *m/z*; stepped normalized collision energy (NCE) 10, 30; dynamic exclusion, 5 sec; top 8 masses selected for MS/MS per scan. Inclusion lists with 20 sec windows were generated in Metaboseek for targeted MS/MS.

**Metaboseek analysis.** HPLC-HRMS data were analyzed using Metaboseek software (documentation available at https://metaboseek.com/) after conversion to mzXML file format using MSConvert (version 3.0, ProteoWizard[56]); for a full list of supported file types, see section 3.3.4 *Supported File Types*. A subset of the mzXML files used in this study are provided as an example data set included with the Metaboseek download at https://metaboseek.com/. The authors recommend installing and running the software locally. For large datasets, 32 GB memory and modern processor (Intel core i7 7700 / Ryzen 7 1700 or better) is recommended. For analysis with up to 50 files (28-minute method), 16 GB of memory are usually enough. Following conversion to mzXML, data were analyzed using the XCMS-module within Metaboseek with default settings, as described in section 3.5 *XCMS Analysis*. Peak detection was carried out with the *centWave* algorithm using the "Metaboseek_default" settings: 4 ppm, 3_20 peakwidth, 3 snthresh, 3_100 prefilter, FALSE fitgauss, 1 integrate, TRUE firstBaselineCheck, 0 noise, wMean mzCenterFun, −0.005 mzdiff. Default settings for XCMS feature grouping: 0.2 minfrac, 2 bw, 0.002 mzwid, 500 max, 1 minsamp, FALSE usegroups. Metaboseek peak filling used the following settings: 3 ppm_m, 3 rtw, TRUE rtrange, FALSE areaMode. The XCMS-generated feature table was loaded into the Metaboseek *Data Explorer* along with relevant MS files by designating a project folder, see section 3.3.2 *Load a Metaboseek Project Folder*. MS data display and MS data table were grouped according to genotype and experimental condition via *Regroup MS data* (section 3.4.2) and *Regroup Table* (section 3.4.4.4), respectively. After defining groups as (i) *C. elegans* samples, (ii) bacterial matrix samples, or (iii) blanks, blank subtraction was performed so that any feature less than ten-fold more abundant in *C. elegans* samples than in blanks was removed. The resulting feature list was further culled using the *Fast Peak Shapes* (Peak Quality) analysis (Supplementary Data 1). These settings were selected for discovery-oriented comparative metabolomics, aiming to retain as many likely "real" features as possible. The resulting table was then regrouped according to *C. elegans* genotype (see *Regroup Table*, section 3.4.4.4) and *Basic Analysis* was performed with WT as the "control" group; see section 3.4.4.3 *Analyze Table* for a complete list of analysis and normalization options. Following analysis, relevant MS/MS data were loaded as described in section 3.3.1 *Load MS Data Files Directly*, followed by matching of MS/MS scans to the MS1 files as described in section 3.4.4.3.3 *Advanced Analysis* under the subheading *Find MS2 scans*. At this point, the table was culled to include only MS/MS-matched features, then culled further using a retention time filter, as described in section 3.4.4 *Feature Table Actions*. MS/MS networking was performed as described in section 3.4.2.2 *Compare MS2*, see also Supplementary Fig. 1 for an example of the *Simplify Network* modifications and display options. All plots and reports, including SIRIUS fragmentation trees, can be exported as vector graphics for rapid and efficient sharing, outlined in section 3.4.2.2 *Feature Report*. HRMS data were analyzed using "Metaboseek_default" settings and normalized to the abundance of ascr#3. Statistical analysis by unpaired two-sided *t*-test was performed with Metaboseek. The identities of all quantified metabolites were verified by analysis of MS/MS spectra and/or synthetic standards.

**Isolation and NMR spectroscopy of bemeth#3 (11) and GLEA-m16:1 (16).** For isolation of GLEA-m16:1 (**16**) from adult *exo*-metabolome, conditioned media from several medium scale *C. elegans hacl-1* cultures was lyophilized and extracted with

methanol (as described). Dried methanol extract was loaded onto Celite and fractionated using medium pressure reverse phase chromatography (15 g C18 Combiflash RediSep column, Teledyne Isco 69-2203-334). Solvent A: 0.1% acetic acid in water; solvent B: acetonitrile. Column was primed with 1% B; separation was achieved by 5% B for 2 column volumes (CV), which was increased linearly to 50% B over 15 CV, then to 100% B over 3 CV and held at 100% B for 5 CV, before returning to 80% B for 3 CV. Fractions were assayed for compounds of interest by HPLC-MS, the relevant fractions were combined and dried in vacuo. Following suspension in water: methanol (1:2), the pooled fractions were further separated by semi-preparative HPLC on a Thermo Hypersil Gold C18 column (250 mm × 10 mm, particle size 5 μM; 25005-259070) using a Vanquish UPLC system controlled by Chromeleon Software (ThermoFisher Scientific) and coupled to a Dionex UltiMate 3000 Automated fraction collector and to an Orbitrap Q-Exactive High Field mass spectrometer using a 9:1 split. Fractions containing GLEA-m16:1 were combined and analyzed by 2D NMR spectroscopy (CD₃OD, Bruker AVANCE III HD, 800 MHz). For spectroscopic data, see Supplementary Table 4. For isolation of bemeth#3, conditioned media from several starved L1 cultures was extracted and fractionated analogously. For spectroscopic data, see Supplementary Table 2, and relevant section of the dqfCOSY spectrum are shown in Supplementary Fig. 4.

**Reporting summary**. Further information on research design is available in the Nature Research Reporting Summary linked to this article.

## Data availability
The HPLC-HRMS data generated during this study have been deposited in the MassIVE database under accession code MSV000087885. Source data are provided with this paper.

## Code availability
Metaboseek is available as an R package, with installation instructions for Windows, macOS, and Linux. A preconfigured R-portable installation is available as installer or.zip file for Windows. Source code, documentation and a tutorial vignette are available at https://github.com/mjhelf/Metaboseek and https://doi.org/10.5281/zenodo.336008757. Individual functions for spectra comparison, merging spectra and filtering molecular formulae have been moved to the companion R package MassTools, with source code and documentation available at https://github.com/mjhelf/MassTools and https://doi.org/10.5281/zenodo.572562058. The Metaboseek tutorial is also available at https://metaboseek.com/doc.

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

## Acknowledgements

This research was supported in part by the National Institutes of Health (R35GM131877 and U01GM110714 to F.C.S.). F.C.S. is a Faculty Scholar of the Howard Hughes Medical Institute. M.J.H received a Research Fellowship from the Deutsche For-schungsgemeinschaft (DFG), Project Number 386228702. Some strains used in this work were provided by the CGC, which is funded by the NIH Office of Research Infrastructure Programs (P40 OD010440). B0334.3(tm6725) was obtained from the National Bior-esource Project, Tokyo, Japan. We thank Diana Carolina Fajardo Palomino and Gary Horvath for technical support and David Kiemle for assistance with NMR spectroscopy. We thank Dr. Yong-Uk Lee and the Walhout lab for kindly sharing the JW1653-1 *E. coli* strain.

## Author contributions

F.C.S. supervised the study. M.J.H. and F.C.S. conceived the Metaboseek platform. M.J.H. developed the Metaboseek platform. B.W.F. and A.B.A. performed chemical and biolo-gical experiments. Y.K.Z. performed syntheses. B.W.F., M.J.H., and F.C.S. wrote the paper with input from all authors.

## Competing interests

The authors declare no competing interests.
