## [Peer Review File · Nature Communications]

REVIEWER COMMENTS

Reviewer #1 (Remarks to the Author):

The manuscript by Helf et al described MetaboSeek for comparative metabolomics. The utility of the tool has been illustrated on a study of a conserved fat metabolism pathway in *C. elegans*. Overall, MetaboSeek fills a nice niche in current metabolomics toolkits by offering a web-based, locally installable, one-stop tool (MS1, isotopes, MS2, structure elucidation) – this is a very useful addition to the community. The interface is well-designed with detailed tutorials and documentations.

My comments are given below:

Comments for MetaboSeek

1) It will be useful to have a progress bar or add some mechanism for status updates. We tested using the example data, and it runs for a very long time. It is not clear when it will finish or it freezes or errors

2) I would suggest to have a more detailed comparison table with other GUI-based tools (such as XCMS online, MS-DIAL, MetaboAnalyst) to illustrate the unique features and advantages of using this tool.

Comments on the case study

3) What is the development period of *hacl-1* mutant (L1 to adult stage)? As mutations in enzyme upstream (PHYH) of the pathway is involved in Refsum's syndrome in humans, it would be interesting to comment on this.

4) In the exo-metabolomics part of the study (Supplementary Figure #8), were the revealed ascarosides by MS1 molecularly characterized using MS2? This is could be an additional benchmark

Comments on the introduction

5) A general LC-HRMS run (full scan mode, MS1 only) gives ~10,000 peaks. This number is quite high, need to specify the context or analytical conditions (including MS2?).

6) "... most of them represent unidentified metabolites.." There is an increasing awareness that most of the peaks could be artifacts. Please refer to "Chemical Discovery in the Era of Metabolomics" (doi: 10.1021/jacs.9b13198). It will be nice to comment on this

Reviewer #2 (Remarks to the Author):

In "Comparative metabolomics with Metaboseek reveals functions of a conserved fat metabolism pathway in *C. elegans*" Helf and coworkers introduce Metaboseek, an open-source software platform for non-targeted metabolomics data analysis. They showcase the utility of metaboseek by exploring a lipid metabolic pathway in *C. elegans*. After comparative metabolomics of starved and fed animals as well as a KO mutant and the wild type, the authors prioritized differential features and clustered them via MS/MS-based molecular networking.

The paper is overall well written and the methods and data representation looks solid. I especially appreciate their effort in following up and confirming the structure of GLEA-m16:1 via NMR and level 1 IDs through in-house synthesized authentic standards. For the MS/MS data I can only judge the exemplary data in the manuscript and SI as the raw data in their massive dataset was not publicly accessible (see comment below).

As you will see in my comments below, I think there is an issue with noise filters/ threshold as the total feature number looks pretty high. I would suggest to inspect this (e.g. with an intensity histogram) and eventually increase the signal threshold. I'm sure this will also have a significant impact on the scalability of the data processing.

On the novelty side, one has to mention that all tools used here (e.g. feature finding, data visualization, adduct annotation, molecular networking, MF annotation) are available as standalone tools and at least in part integrated in other platforms (e.g. XCMSOnline, GNPS, Galaxy, Compound Discoverer, EIMaven). Nevertheless, I think that there is still a need for simple to use and fully integrated metabolomics data analysis platforms, and in combination with the appealing biological case study the paper should be of broad interest for the metabolomics community and biologist who want to apply non-targeted metabolomics to their particular system.

What I would encourage the authors to do is to discuss the overlap and differences to other exciting metabolomics data analysis tools (many of them are cited already) more in detail, and perhaps point out more clearly what gaps still exist, and which ones can be overcome with metaboseek.

Also, a short overview of the user numbers (e.g. from google analytics from your web server, or download numbers of the code) would be good to get an idea about the level of implementation of the tool in the community so far.

In order to judge the utility of metaboseek, I would also encourage the authors to provide a short statement on their server capacity and scalability, as well as hardware requirements for the offline version (e.g. What do I need and how long does it take to process, 10; 100; 1,000 and 10,000 LC-MS/MS files).

Detailed comments:

Line 39: Consider indicating the abbreviation HPLC

Line 40: 100,000 features sounds a little extreme. Is this before blank subtraction and noise filtering?

Line 14-15 and 42: I don't think that "most" of those (several 100,000) features are unknown compounds but rather noise, different adducts and other artifacts of some unknown compounds. This is certainly an ongoing discussion in the field, so perhaps expand the discussion a little here and clarify whether you really expect hundreds of thousands of new compounds within those features.

Line 50: Should be "XCMS" and "MZmine" (perhaps check this throughout the paper)

Line 52: Consider mentioning other open databases (e.g. Massbank) as well.

Line 55: Consider indicating the abbreviation "MS/MS".

Line 57-59: While I agree with the statement in general, I wonder what aspects what particular functions are needed, that are not provide in other metabolomics platforms. Perhaps a short explanation would set your needs better in perspective of existing tools (E.g. XCMSonline, Galaxy, GNPS Dashboard, EIMaven, Compound Discoverer etc.)

Line 66-68: How do you integrate the annotated adducts into your network? By annotation alone, or also through additional network connections (see R Schmid et al Nat Com 2021)?

Line 110: Maybe mention through which server metaboseek is available (provide url) and also what capabilities this server has. (e.g. user data storage, scalability).

Line 112: I'd provide the url for the documentation.

Line 202: See my comments above regarding feature numbers. I think this should be revised with more stringent noise cut-off, or discussed why you used such loose settings.

Line 235 and throughout the paper: I would avoid the term "daughter ion" and use "product ion" instead.

Line 337-338: Is there an overview of the tool you can refer too? Or perhaps you could list them in an SI table? I think this would be very helpful for new users.

Data availability: Thanks a lot for uploading your MS data to a public repository. However, I could not access it during review. Perhaps you still need to make it "public". (there is a button in massive you have to click so it becomes part of the public data).

Daniel Petras

Reviewer #3 (Remarks to the Author):

Review Schroeder Nature Communications 2021

General comments:

This is a very nice paper introducing a novel tool (Metaboseek) to analyze and interpret LC-MS metabolomics data. I am not an LC-MS analysis specialist and will leave it to the other reviewers to comment on how well the Metaboseek tool works, especially relative to other tools (where available).

Overall, I enjoyed reading this paper and I think the work has high potential impact for the metabolomics community. As written, I found it somewhat schizophrenic though as it flips from tool building to biology and back. I suggest the authors first present the tool – what is it, what of known resources/tools does it combine, where is it available, can it handle large datasets, how is a user to use the tool, i.e., via a website or run locally, trouble shooting, pros and cons, etc.

Then follow with the example of hacl-1 mutants, but be careful not to overstate implications for the entire pathway and/or relevance to humans (see below). Specifically, “the metaboseek workflow” section is a bit tough to read since I am not an expert. However, wouldn’t it be fabulous if a non-expert like myself could generate LC-MS data and then have the ease of using a tool like this? I think it would be great if the authors could write the paper such that it appeals to and is useful for both high-level LC-MS users like they are and for biologists like myself.

Minor comments:

- When comparing two conditions or strains, how are the data normalized to make sure that differences are due to condition/strain rather than amount of input material?

- It would be helpful to have a cartoon of the hacl-1 gene/protein and the mutant used. The allele should also be mentioned in the text and figures. Also, while similarity of ZK550.5 and ZK550.6 to the human enzymes is mentioned, nothing is mentioned about the homology of HACL-1 to its human ortholog.

- I think it would be great to have all metabolites that differ in hacl-1 mutants at either developmental stage in a single figure so readers can judge which are specifically different in which stage. A broader discussion about why these differences have such a strong developmental bias would be useful as well.
- I strongly object to using the term “upregulated” for a metabolite that is in high abundance in one strain versus another. When the enzyme required to break down a metabolite is missing (here HACL-1), metabolites (substrates or metabolites derived thereof) simply accumulate. “Regulate” has a different connotation in my opinion.
- The section on the five subnetworks was hard to read.
- I appreciate the experiments showing that the unsaturated odd chain fatty acids do not originate from BCFA metabolism.
- The discussion was rather disjointed, hopping from the tool to the biology back to the tool back to the biology. The second paragraph could be removed. I suggest a strong and organized discussion about the tool – who is it useful for, what are the strengths and limitations.

Specific comments:

- Page 2, line 18: “elucidate functions of a conserved fat metabolism pathway” – I think elucidate functions is vague especially since no physiological studies beyond metabolite level measurements are done. I also don’t think they study the pathway, just one gene, but that is a very minor point the authors could elect to ignore.
- Page 2, line 26/27. Remove the reference to human diets. I understand the pressure the authors may feel to have the study be relevant to humans, but I strongly disagree – this work is potentially impactful for a whole host of researchers – both technical and biological, and the ‘must have human relevance’ is a dangerous meme indeed.
- Page 3, line 64. Figure 1 does not show a browser-based graphical user interface. In fact, it would be great if it would show that.
- Page 3, line 69. In my opinion, there is no need to abbreviate peroxisomal β -oxidation – I find it awkward in reading.
- Page 3, line 71. Change “the metabolism of ...” to “the breakdown of...”
- Page 4, line 77. Provide reference for the statement that *C. elegans* β -oxidation has been investigated.
- Page 4, line 82. Remove “which are also part of human diets”, which reads like an afterthought – see above.
- Page 5, line 129. What is the evidence that the hacl-1 mutant is a null?

- Page 7, lines 193-197. These last two sentences are incomprehensible to me.
- Page 11, line 302. No need to refer to a 'model' – simply say something like “our data suggest that diet-derived.... Fatty acids are initially chain sorted by β -oxidation...”
- Page 27, figure 2A. Shouldn't β -oxidation be a cycle until a fatty acid suitable for β -oxidation is generated?
- Page 28 and 29 – the MS/MS networks are hard to look at and could be explained better.

I hope these comments and suggestions help the authors to further improve their paper.

Marian Walhout

Response to the reviewers' comments.

Reviewer #1 (Remarks to the Author):

The manuscript by Helf et al described MetaboSeek for comparative metabolomics. The utility of the tool has been illustrated on a study of a conserved fat metabolism pathway in *C. elegans*. Overall, MetaboSeek fills a nice niche in current metabolomics toolkits by offering a web-based, locally installable, one-stop tool (MS1, isotopes, MS2, structure elucidation) – this is a very useful addition to the community. The interface is well-designed with detailed tutorials and documentations.

My comments are given below:

Comments for MetaboSeek

1) It will be useful to have a progress bar or add some mechanism for status updates. We tested using the example data, and it runs for a very long time. It is not clear when it will finish or it freezes or errors

Response: We thank the reviewer for this excellent suggestion. In the newest release (0.9.9.1, now available for download at metaboseek.com), we added progress bars for processes with longer runtimes (e.g., loading of MS data files, “Fast Peak Shapes” peak scoring, MS/MS Networking, and Label Finder). In addition, any multithreaded process will now show a progress bar in the R console window which can be useful when running Metaboseek locally on a workstation, which is the preferred usage scenario. For monitoring progress of the XCMS analysis, there is a progress window at the bottom of the XCMS analysis module that updates as analysis steps are completed. It is also possible to open the status.csv file in the results folder to view progress (when using Linux).

These progress bar screenshots were taken during unbiased isotope tracing, separated by 10 minutes (total task time ~2 hours). Prior processing of the 4 file example data set provided at metaboseek.com takes about 1.5 h, including XCMS analysis, RT correction, and CAMERA assignment, using a fairly standard Intel(R) Core(TM) i7-9750H CPU @ 2.60GHz RAM: 32GB, with two workers (cores) enabled.

2) I would suggest to have a more detailed comparison table with other GUI-based tools (such as XCMS online, MS-DIAL, MetaboAnalyst) to illustrate the unique features and advantages of using this tool.

Response: We thank the reviewer for this suggestion and have compiled a comparison table for GUI-based metabolomics analysis tools and added it to the manuscript. See **Supplementary Table S1**.

Comments on the case study

3) What is the development period of *hacl-1* mutant (L1 to adult stage)? As mutations in enzyme upstream (PHYH) of the pathway is involved in Refsum's syndrome in humans, it would be interesting to comment on this.

Response: We appreciate the reviewer's question and have included in the text that developmental pace of *hacl-1* mutants is similar to N2 animals, see Line 141.

4) In the exo-metabolomics part of the study (Supplementary Figure #8), were the revealed ascarosides by MS1 molecularly characterized using MS2? This is could be an additional benchmark

Response: Identities of all ascarosides were confirmed by MS2. We added a comment in methods to clarify this point.

Comments on the introduction

5) A general LC-HRMS run (full scan mode, MS1 only) gives ~10,000 peaks. This number is quite high, need to specify the context or analytical conditions (including MS2?).

Response: In our experience, injection of a concentrated biological extract (e.g., *C. elegans*-derived) can result in detection of several 100,000 MS1 features when using low signal-to-noise thresholds in XCMS, as suited for discovery-focused metabolomics (see Methods for XCMS parameters). After feature detection, we performed blank subtraction by grouping all worm-derived samples and imposing a foldchange threshold over blank samples. This list was further culled using a peak quality threshold, see Lines 441-450 and new **Supplementary Table S6**.

6) "... most of them represent unidentified metabolites.." There is an increasing awareness that most of the peaks could be artifacts. Please refer to "Chemical Discovery in the Era of Metabolomics" (doi: 10.1021/jacs.9b13198). It will be nice to comment on this

Response: This is an important point and we have revised the text to clarify the distinction between numbers of features and numbers of metabolites, e.g., see lines 41-43 "... can reveal more than 100,000 molecular features (defined by a mass-to-charge ratio, m/z , and retention time, RT), representing a complex mixture of ions derived from known and unknown metabolites, adducts, naturally occurring isotopes and background."

Several recent studies have shown that, generally, numbers of detected features are at least one order of magnitude higher than numbers of compounds, and that comprehensive adduct annotation remains a major challenge. The current version of Metaboseek integrates CAMERA analysis for basic adduct assignments, and we are looking to integrate more advanced adduct assignment tools in the future, e.g., as implemented in a recent study by Mahieu and Patti (doi: 10.1021/acs.analchem.7b02380).

Nonetheless, in our experience, the number of yet unidentified metabolites in *C. elegans* and other model organisms remains high compared to the number of known compounds. For example, many of the unknowns at intermediate polarity appear to represent modular metabolites that are combinations of primary metabolic building blocks, e.g., the large number of previously undescribed lipid derivatives in the current study, or the modular glucosides, several 100 of which we partially characterized, see Artyukhin et al. JACS 2018 (doi: 10.1021/jacs.7b11811), Wrobel et al. JACS 2021 (doi: 10.1021/jacs.1c05908), also see Chang F-Y et al. Nature Microbiology 2021 (doi: 10.1038/s41564-021-00887-y) as an example for combinatorial assembly of amino acids and fatty acids.

Reviewer #2 (Remarks to the Author):

In "Comparative metabolomics with Metaboseek reveals functions of a conserved fat metabolism pathway in C. elegans" Helf and coworkers introduce Metaboseek, an open-source software platform for non-targeted metabolomics data analysis. They showcase the utility of metaboseek by exploring a lipid metabolic pathway in C. elegans. After comparative metabolomics of starved and fed animals as well as a KO mutant and the wild type, the authors prioritized differential features and clustered them via MS/MS-based molecular networking.

The paper is overall well written and the methods and data representation looks solid. I especially appreciate their effort in following up and confirming the structure of GLEA-m16:1 via NMR and level 1 IDs through in-house synthesized authentic standards. For the MS/MS data I can only judge the exemplary data in the manuscript and SI as the raw data in their massive dataset was not publicly accessible (see comment below).

As you will see in my comments below, I think there is an issue with noise filters/ threshold as the total feature number looks pretty high. I would suggest to inspect this (e.g. with a intensity histogram) and eventually increase the signal threshold. I'm sure this will also have a significant impact on the scalability of the data processing.

Response: We appreciate the reviewer's perspective on filters/threshold levels, and we have revised the text to estimate numbers of features in metabolomics studies more conservatively. Further, we have added **Supplementary Table S6** listing total number of features detected, with and without blank subtraction, using different Peak Quality thresholds.

Metaboseek provides the ability to quickly sort data based on user-defined criteria, e.g., intensity or foldchange, facilitating management of large datasets. Generally, we believe that retaining as many features as reasonably possible throughout the initial stages of the analysis is advantageous, although we agree with the reviewer's point that increasing noise thresholds can be necessary when scaling to larger datasets.

On the novelty side, one has to mention that all tools used here (e.g. feature finding, data visualization, adduct annotation, molecular networking, MF annotation) are available as standalone tools and at least in part integrated in other platforms (e.g. XCMSonline, GNPS, Galaxy, Compound Discoverer, EIMaven). Nevertheless, I think that there is still a need for simple to use and fully integrated metabolomics data analysis platforms, and in combination with the appealing biological case study the paper should be of broad interest for the metabolomics community and biologist who want to apply non-targeted metabolomics to their particular system. What I would encourage the authors to do is to discuss the overlap and differences to other exciting metabolomics data analysis tools (many of them are cited already) more in detail, and perhaps point out more clearly what gaps still exist, and which ones can be overcome with Metaboseek.

Response: We agree that a more detailed comparison is useful, and have added an overview table of different available MS metabolomics tools with a focus on those platforms built around a GUI, see **Supplementary Table S1**. In addition, we expanded the discussion to summarize the advantages Metaboseek confers for discovery-oriented metabolomics.

Also, a short overview of the user numbers (e.g. from google analytics from your web server, or download numbers of the code) would be good to get an idea about the level of implementation of the tool in the community so far.

Response: One of the advantages of Metaboseek is that it allows running analyses locally, on the user's computer. In most usage scenarios, this is much faster and more convenient than interacting with the online version, and thus Metaboseek is used mostly offline. The offline version does show the 'news' section of the Metaboseek website with information about the latest available version on the app start page. The number of accesses to the news page can therefore give us (likely incomplete) usage information. According to Google analytics, there were 392 individual user sessions of the app in the last three months (either in the online version or running locally) after removing known bots.

In order to judge the utility of metaboseek, I would also encourage the authors to provide a short statement on their server capacity and scalability, as well as hardware requirements for the offline version (e.g. What do I need and how long does it take to process, 10; 100; 1,000 and 10,000 LC-MS/MS files).

Response: We thank the reviewer for this helpful suggestion. Most users run the software locally for up to several hundred LC-MS/MS files. For large datasets, 32 GB memory and fast processor (Intel core i7 7700 / Ryzen 7 1700 or better) is recommended. For analyses with up to 50 files (e.g., 28 min HPLC, m/z 100-1000), 16 GB of memory are usually enough. These recommendations have been added to the Methods section of the manuscript, see Line 433.

For on-line “try out” of Metaboseek we are currently providing a relatively low spec server to run an example dataset for demonstration purposes but are in conversations with the community to set up a much more powerful web server that can access public datasets from GNPS directly. Further, any user can set up Metaboseek on an online server, which can be configured through shiny server (free for academic users). Like other shiny apps, it is also straightforward to make Metaboseek available on a local network.

Detailed comments:

Line 39: Consider indicating the abbreviation HPLC. Response: Done.

Line 40: 100,000 features sounds a little extreme. Is this before blank subtraction and noise filtering?

Response: In the current study, XCMS peak picking yielded over 500k features in the ESI- and ESI+ runs combined, of which 59k (ESI-) and 106k (ESI+) features remained after blank subtraction and use of a Peak Quality filter (see **Supplementary Table S6** for details). Nonetheless, we agree that not all types of metabolome samples will exhibit such large feature numbers, especially when more stringent peak picking criteria or noise thresholds are used, and therefore we have revised the text to estimate numbers of features in metabolomic studies more conservatively, e.g., see lines 41-43.

Line 14-15 and 42: I don't think that “most” of those (several 100,000) features are unknown compounds but rather noise, different adducts and other artifacts of some unknown compounds. This is certainly an ongoing discussion in the field, so perhaps expand the discussion a little here and clarify whether you really expect hundreds of thousands of new compounds within those features.

Response: Please see our response to a similar comment by reviewer#1. As explained above, we have revised the text to estimate numbers of features in metabolomic analyses more conservatively and clarify the distinction between numbers of features and numbers of metabolites. However, at least in *C. elegans*, many peaks in fact represent unknown compounds, and thus we believe our revised wording that “many of which may represent unknowns” is appropriate.

Line 50: Should be “XCMS” and “MZmine” (perhaps check this throughout the paper)

Response: Corrected, thanks!

Line 52: Consider mentioning other open databases (e.g. Massbank) as well.

Response: Thank you for the suggestion. We have added Massbank as an additional example.

Line 55: Consider indicating the abbreviation “MS/MS”. Response: Done.

Line 57-59: While I agree with the statement in general, I wonder what aspects what particular functions are needed, that are not provide in other metabolomics platforms. Perhaps a short explanation would set your needs better in perspective of existing tools (E.g. XCMSonline, Galaxy, GNPS Dashboard, EIMaven, Compound Discoverer etc.)

Response: We appreciate this point and have added a comparison table to clearly outline capabilities of other free and open-source metabolomics software that we believe to be the most relevant reference points (**Supplementary Table S1**).

Line 66-68: How do you integrate the annotated adducts into your network? By annotation alone, or also through additional network connections (see R Schmid et al Nat Com 2021)?

Response: Adducts were included in the MS/MS networks (e.g., Figure 3a), in some cases resulting in redundant subnetworks for sodium formate adducts (see below). It may prove useful to integrate the IIMN package into Metaboseek, we thank the reviewer for this suggestion.

Line 110: Maybe mention through which server metaboseek is available (provide url) and also what capabilities this server has. (e.g. user data storage, scalability).

Response: Most users run the software locally on their own computers for up to several hundred LC-MS/MS files. At present, we are only providing an example app, currently at mosaic.bti.cornell.edu/Metaboseek – the link to this app is on Metaboseek.com. As stated above, we are in conversations with the community to set up a more powerful web server that allows more direct access to datasets on the GNPS/MassIVE servers.

Line 112: I'd provide the url for the documentation.

Response: Thanks for the suggestion. We added the URL for the Metaboseek website including documentation and installation files.

Line 202: See my comments above regarding feature numbers. I think this should be revised with more stringent noise cut-off, or discussed why you used such loose settings.

Response: We have revised the methods to include more detail on the specific settings used, which aim to retain as many likely “real” features as possible for the downstream comparative analysis, see Lines 470-500 (“Metaboseek Analysis”). Importantly, Metaboseek enables flexible and quick filtering of data *after* feature detection and thus facilitates using a low noise cut-off; initially retaining as many features as possible does not burden downstream analysis.

Line 235 and throughout the paper: I would avoid the term “daughter ion” and use “product ion” instead. **Response:** Done.

Line 337-338: Is there an overview of the tool you can refer to? Or perhaps you could list them in an SI table? I think this would be very helpful for new users.

Response: Thank you for the suggestion. More detailed documentation for each individual module of the Metaboseek R package is available online at <https://rdrr.io/github/mjhelf/Metaboseek/man/>. We have added a reference to the general documentation to guide interested readers to this resource.

Data availability: Thanks a lot for uploading your MS data to a public repository. However, I could not access it during review. Perhaps you still need to make it “public”. (there is a button in massive you have to click so it becomes part of the public data).

Response: We apologize for the oversight. The data has now been made public. Thank you for bringing this to our attention!

Reviewer #3 (Remarks to the Author):

This is a very nice paper introducing a novel tool (Metaboseek) to analyze and interpret LC-MS metabolomics data. I am not an LC-MS analysis specialist and will leave it to the other reviewers to comment on how well the Metaboseek tool works, especially relative to other tools (where available).

Overall, I enjoyed reading this paper and I think the work has high potential impact for the metabolomics community. As written, I found it somewhat schizophrenic though as it flips from tool building to biology and back. I suggest the authors first present the tool – what is it, what of known resources/tools does it combine, where is it available, can it handle large datasets, how is a user to use the tool, i.e., via a website or run locally, trouble shooting, pros and cons, etc.

*Then follow with the example of *hacl-1* mutants, but be careful not to overstate implications for the entire pathway and/or relevance to humans (see below). Specifically, “the metaboseek workflow” section is a bit tough to read since I am not an expert. However, wouldn’t it be fabulous if a non-expert like myself could generate LC-MS data and then have the ease of using a tool like this? I think it would be great if the authors could write the paper such that it appeals to and is useful for both high-level LC-MS users like they are and for biologists like myself.*

Response: Thank you for the helpful suggestions. To avoid flipping back and forth from tool building to biology, we have consolidated paragraphs on MS/MS and other technical aspects to the “Workflow” section. In addition, we have moved some technical terms and the list of other software tools to **Supplementary Table S1**.

Minor comments:

- When comparing two conditions or strains, how are the data normalized to make sure that differences are due to condition/strain rather than amount of input material?

Response: This is an excellent question and an ongoing discussion in the metabolomics community. Metaboseek offers several different normalization options, as described in the documentation; however, in most cases selection of the appropriate parameters depends on specific characteristics of the analysis and requires user input. For example, in the present study we used the ascaroside *ascr#3*, which is produced abundantly at all *C. elegans* life stages, as a “housekeeping” metabolite, reflecting total number of animals in each sample. In other cases, the comparison of ratios of metabolites in different mutants (or under different conditions) may be most appropriate.

*- It would be helpful to have a cartoon of the *hacl-1* gene/protein and the mutant used. The allele should also be mentioned in the text and figures. Also, while similarity of ZK550.5 and ZK550.6 to the human enzymes is mentioned, nothing is mentioned about the homology of *HACL-1* to its human ortholog.*

Response: We thank the reviewer for these suggestions and have included a cartoon of the gene and deletion in Figure 2. We have added sentences to describe the similarity of *hacl-1* to the previously

characterized human gene (see Line 134), and to more clearly define the gene disruption in the mutant (see Line 140).

- I think it would be great to have all metabolites that differ in hac1-1 mutants at either developmental stage in a single figure so readers can judge which are specifically different in which stage. A broader discussion about why these differences have such a strong developmental bias would be useful as well.

Response: We appreciate the reviewer's suggestion and expanded **Supplementary Figure 6** to include intensity values for C₁₁ compounds in both larvae and adults on the same graph. This class of compounds was strongly enriched in larvae and also detected in adults (although not as abundant nor as differential in adults). As explained in the text, the vast majority of differential metabolites related to fatty acid and ethanolamide metabolism were detected robustly only in adults, likely from the consumption of bacterial lipids.

- I strongly object to using the term "upregulated" for a metabolite that is in high abundance in one strain versus another. When the enzyme required to break down a metabolite is missing (here HAC1-1), metabolites (substrates or metabolites derived thereof) simply accumulate. "Regulate" has a different connotation in my opinion.

Response: We thank the reviewer for highlighting this distinction, and we fully agree. We have replaced the word "upregulated" with "enriched" to convey that these compounds are accumulating in the mutant.

- The section on the five subnetworks was hard to read.

Response: We revised this section to make it more accessible, especially the parts dealing with interpretation of MS/MS spectra. Technical parts have been moved to the initial "Workflow" section.

- I appreciate the experiments showing that the unsaturated odd chain fatty acids do not originate from BCFA metabolism.

Response: Thanks for the comment. We felt it is important to clearly demonstrate that BCFA metabolism is separate.

- The discussion was rather disjointed, hopping from the tool to the biology back to the tool back to the biology. The second paragraph could be removed. I suggest a strong and organized discussion about the tool – who is it useful for, what are the strengths and limitations.

Response: We appreciate the reviewer's suggestion and expanded the discussion of the utility of Metaboseek for discovery, facilitating structure elucidation and discovery of biosynthetic networks and functional context of new metabolites. We also shortened and reorganized the second paragraph (now the first paragraph) in the discussion to avoid repetition of results.

Specific comments:

- Page 2, line 18: "elucidate functions of a conserved fat metabolism pathway" – I think elucidate functions is vague especially since no physiological studies beyond metabolite level measurements are done. I also don't think they study the pathway, just one gene, but that is a very minor point the authors could elect to ignore.

Response: We thank the reviewer for raising this issue and have modified the text.

- Page 2, line 26/27. Remove the reference to human diets. I understand the pressure the authors may feel to have the study be relevant to humans, but I strongly disagree – this work is potentially impactful for a whole host of researchers – both technical and biological, and the 'must have human relevance' is a dangerous meme indeed.

Response: We have removed the reference to human diets from the abstract, per the reviewer's suggestion.

- Page 3, line 64. Figure 1 does not show a browser-based graphical user interface. In fact, it would be great if it would show that.

Response: We now include a link to [Metaboseek.com](https://metaboseek.com) which features a full-page view of the Metaboseek user interface.

- Page 3, line 69. In my opinion, there is no need to abbreviate peroxisomal α -oxidation – I find it awkward in reading.

Response: In the past we have used similar abbreviations, e.g., p β o and m β o, to represent peroxisomal and mitochondrial β -oxidation, respectively. However, we are open to removing the abbreviation, as suggested by the reviewer. We will defer to the editor's opinion.

- Page 3, line 71. Change “the metabolism of ...” to “the breakdown of...” **Response:** Done.

- Page 4, line 77. Provide reference for the statement that *C. elegans* β -oxidation has been investigated.

Response: There is a reference for this statement, see Artyukhin et al., JACS 2018 (doi: 10.1021/jacs.7b11811).

- Page 4, line 82. Remove “which are also part of human diets”, which reads like an afterthought – see above. **Response:** We removed the clause as suggested by the reviewer.

- Page 5, line 129. What is the evidence that the *hacl-1* mutant is a null?

Response: We have modified the text to more clearly describe the mutation and included a cartoon of the gene and deletion in Figure 2. The allele *tm6725* carries a 408 bp deletion ablating an exon splice junction; the non-spliced mRNA, if not degraded, would produce a truncated protein missing conserved functional domains.

- Page 7, lines 193-197. These last two sentences are incomprehensible to me.

Response: We revised this section to make it clearer.

- Page 11, line 302. No need to refer to a ‘model’ – simply say something like “our data suggest that diet-derived... Fatty acids are initially chain sorted by β -oxidation...”

Response: We simplified the sentence as suggested.

- Page 27, figure 2A. Shouldn't α -oxidation be a cycle until a fatty acid suitable for β -oxidation is generated?

Response: One complete iteration of α -oxidation is described in the figure, in which a β -methyl fatty acid is chain-shortened by one carbon, yielding an α -methyl fatty acid that can be further degraded by β -oxidation.

- Page 28 and 29 – the MS/MS networks are hard to look at and could be explained better.

Response: We added a more detailed explanation of MS/MS networking in a dedicated paragraph in the “Workflow section.”

I hope these comments and suggestions help the authors to further improve their paper.

Response: We greatly appreciate the many excellent suggestions!

REVIEWERS' COMMENTS

Reviewer #2 (Remarks to the Author):

Thanks a lot for addressing my comments and for making the data public. Overall, I'm satisfied with the response and the revised manuscript and the raw data I inspected looks solid.

One thing the authors might still want to consider, is describing the different feature numbers and effects of the filtering from table S6. I assume a short statement in the main text, on how blank subtraction and peak quality thresholds effected the overall feature numbers would be quiet interesting for many readers. An interesting aspect could also be how these cut-offs changed spectral annotation rates.

Other than this, the manuscript seems to be ready for publication and I'm looking forward to see it in its printed form.

Response to the reviewers' comments

Reviewer #2 (Remarks to the Author):

Thanks a lot for addressing my comments and for making the data public. Overall, I'm satisfied with the response and the revised manuscript and the raw data I inspected looks solid.

One thing the authors might still want to consider, is describing the different feature numbers and effects of the filtering from table S6. I assume a short statement in the main text, on how blank subtraction and peak quality thresholds effected the overall feature numbers would be quiet interesting for many readers. An interesting aspect could also be how these cut-offs changed spectral annotation rates.

Other than this, the manuscript seems to be ready for publication and I'm looking forward to see it in its printed form.

Response: We thank the reviewer for his positive feedback on the revised manuscript. As suggested, we have included a brief description on the effects of filtering in the main text (see Lines 205-206).